# Radiofrequency Combined with Intratumoral Immunotherapy: Preclinical Results and Safety in Metastatic Colorectal Carcinoma

**DOI:** 10.3390/pharmaceutics16030315

**Published:** 2024-02-23

**Authors:** Johanne Seguin, Mostafa El Hajjam, Josette Legagneux, Sarah Diakhaby, Nathalie Mignet, Vincent Boudy, Balthazar Toussaint, Frederique Peschaud, Jean François Emile, Claude Capron, Robert Malafosse

**Affiliations:** 1Université Paris Cité, CNRS, INSERM, UTCBS, 4 Avenue de l’Observatoire, 75006 Paris, France; johanne.seguin@u-paris.fr (J.S.); sarah.diakhaby@u-paris.fr (S.D.); nathalie.mignet@parisdescartes.fr (N.M.); vincent.boudy@aphp.fr (V.B.); 2Department of Interventional Radiology, Assistance Publique—Hôpitaux de Paris (AP-HP), Ambroise Pare Hospital, 9 Avenue Charles de Gaulle, 92104 Boulogne-Billancourt, France; mostafa.elhajjam@aphp.fr; 3Surgery School AP-HP, 75013 Paris, France; josette.legagneux@aphp.fr; 4IMISCA Therapeutics, 110 Avenue Pierre Brossolette, 92240 Malakoff, France; 5Pharmaceutical Research and Development Department, AGEPS, AP-HP, 75013 Paris, France; balthazar.toussaint@aphp.fr; 6Department of Digestive and Oncologic Surgery, Assistance Publique-Hôpitaux de Paris (AP-HP), Ambroise Pare Hospital, 9 Avenue Charles de Gaulle, 92104 Boulogne-Billancourt, France; frederique.peschaud@aphp.fr; 7EA4340 BCOH, Paris-Saclay University, 91190 Gif sur Yvette, France; jean-francois.emile@uvsq.fr (J.F.E.); claude.capron@aphp.fr (C.C.); 8Pathology Department, Assistance Publique-Hôpitaux de Paris (AP-HP), Ambroise-Pare Hospital, 9 Avenue Charles de Gaulle, 92104 Boulogne-Billancourt, France; 9Immunology Department, Assistance Publique-Hôpitaux de Paris (AP-HP), Ambroise-Pare Hospital, 9 Avenue Charles de Gaulle, 92104 Boulogne-Billancourt, France

**Keywords:** radiofrequency, colorectal cancer, liver metastases, intratumoral immunotherapy, abscopal effect

## Abstract

Radiofrequency ablation (RFA) of cancer induces an anti-tumor immunity, which is insufficient to prevent recurrences. In mice, RFA–intratumoral immunotherapy by granulocyte–macrophage colony-stimulating factor (GM-CSF) and Bacillus Calmette-Guerin resulted in complete metastases regression. Infectious risk in human needs replacement of live vaccines. Intratumoral purified protein derivatives (PPD) have never been tested in digestive cancers, and the safety of intratumoral immunotherapy after RFA has not yet been validated in human models. We investigated the therapeutic efficacy of combined radiofrequency ablation (RFA) and intratumoral immunotherapy (ITI) using an immune-muco-adherent thermogel (IMT) in a mouse model of metastatic colorectal cancer (CRC) and the safety of this approach in a pig model. Intratumoral stability of the immunogel was assessed using magnetic resonance imaging (MRI) and bioluminescent imaging. Seventy-four CT26 tumor-bearing female BALB/c mice were treated with RFA either alone or in combination with intratumoral IMT. Regression of distant metastasis and survival were monitored for 60 days. Six pigs that received liver radiofrequency and intralesional IMT injections were followed for 15 days. Experimental gel embolisms were treated using an intravascular approach. Pertinent rheology of IMT was confirmed in tumors, by the signal stability during 3 days in MRI and 7 days in bioluminescence imaging. In mice, the abscopal effect of RFA–intratumoral immunotherapy resulted in regression of distant lesions completed at day 16 vs. a volume of 350 ± 99.3 mm^3^ in the RFA group at day 25 and a 10-fold survival rate at 60 days. In pigs, injection of immunogel in the liver RFA area was safe after volume adjustment without clinical, hematological, and liver biology disorder. Flow cytometry showed an early increase in CD3 TCRγδ+T cells at D7 (*p* < 0.05) and a late decrease in CD29^+^-CD8 T cells at D15 (*p* < 0.05), reflecting the inflammation status changes. Systemic GM-CSF release was not detectable. Experimental caval and pulmonary thermogel embolisms were treated by percutaneous catheterism and cold serum infusion. RFA–intratumoral immunotherapy as efficient and safe mini-invasive interventional oncology is able to improve ablative treatment of colorectal liver metastases.

## 1. Introduction

Radiofrequency ablation (RFA), the most widely used technology for thermal ablation, extends indications in cancer curative strategies. In unresectable liver metastases of colorectal cancer (CRC), a combination of radiofrequency (RFA) and chemotherapy improves survival compared with chemotherapy alone. However, 80% of the patients who undergo complete macroscopic treatment develop recurrences [1,2,3,4]. Perioperative chemotherapy does not increase the overall survival [5]. Systemic immunotherapies are efficient in several cancers but not in microsatellite-stable digestive cancers. RFA, as non-viral oncolysis, reduces tumor volume, provides specific antigens, and primes a specific immune response [6,7,8,9], which is inadequate to avoid recurrences related to distant, uncontrolled occult microscopic lesions. No study has demonstrated an enhancement of tumor-infiltrating lymphocytes (TILs) in metastases distant from an RFA area. RFA alone is not sufficient to reset the harmful relationship between malignant cells, the tumor micro-environment (TME), and immune cells in distant lesions [10]. The first clinical trials of immunotherapy in CRC used Bacillus Calmette-Guerin (BCG) [11]. In previous preclinical studies, RFA and in situ administration of an immune-muco-adherent thermogel (IMT) containing macrophage colony-stimulating factor (GM-CSF) and BCG treated distant microscopic lesions in mice through a strong specific immune response that induced an abscopal effect. The immune escape of bulky lesions was reversible with the addition of a systemic PD-1 blockade [10,12]. Because BCG immunotherapy induces sepsis [13], live vaccines must be replaced with a safe adjuvant. The efficacy of purified protein derivatives (PPD) of Mycobacterium tuberculosis as pathogen-associated molecular patterns (PAMP) and a toll-like receptor (TLR) agonist [14,15,16] has not been established in digestive cancers.

The purpose of this preclinical study was to examine the efficiency of RFA–intratumoral immunotherapy using GM-CSF and PPD combined in a thermogel to optimize their in situ bioavailability in a murine CRC model. In view of clinical trials, it was essential to explore the safety of this combination in a large animal model.

## 2. Materials and Methods

### 2.1. Animals and Cells

Animal studies, conducted in female immunocompetent syngeneic BALB/c mice (Janvier Labs, FR (Janvier Laboratories; Le Genest Saint l’Isle, France) and female pigs (Limosin FR (Limosin, Neuve Maison, France) in good laboratory practice conditions, were approved by the French Ethics Committees (APAFIS #11352 for mice; authorization no. 37954-2022032912268622. v5 for pig).

Luciferase-expressing cells (CT26-Luc) were produced by transfecting the wt-CT26 cell line (ATCC, CRL-2638, LGC Standards) with the luciferase reporter gene for bioluminescent tumor growth monitoring as largely described in previous studies [17]. This cell line was cultured in Dulbecco’s Modified Eagle Medium (DMEM, Gibco Life Technologies Thermo Fisher, Asnière, France) containing 10% fetal bovine serum (FBS, Gibco Life Technologies, Thermo Fisher, Asnière, France), 100 μM streptomycin, 100 U/mL penicillin, and 0.4 mg/mL geneticin (G418 sulfate, Gibco Life Technologies Thermo Fisher, Asnière, France) at 37 °C in a humidified atmosphere of 5% CO_2_.

For establishing a well-standardized primitive tumor model, a subcutaneous CT26 tumor-bearing mouse was sacrificed and the lesion resected and divided into 30 mm^3^ fragments. For magnetic resonance imaging (MRI), optical imaging, and RFA efficacy evaluation, these tumor fragments were inserted subcutaneously with a 12-gauge trocar (38 mm) into the flank of each mouse. Seventy-four six-week-old female BALB/c mice, used to assess the efficacy of PPD in intratumoral immunotherapy (ITI), were anesthetized by ketamine (100 mg/kg), xylazine (10 mg/kg), or 2.5% inhaled isoflurane. Six 10-week-old female domestic pigs (30–32 kg), used as models to mimic human hepatic and cardiopulmonary anatomy, were anesthetized by intramuscular ketamine (20 mg/kg), intravenous propofol (3.3 mg/kg), and 2.5% inhaled isoflurane. 

### 2.2. Immune-Muco-Adherent Thermogel Preparation and Characterization

A stock solution of poloxamer P407/xanthan gum (XG) 25/0.12% *w*/*v* (Kolliphor^®^P407; Satiaxane CX 930, BASF, Boussens, France) was prepared in sterile water [18]. In a preliminary study, three PAMP candidates combined with GM-CSF in IMT (n = 30) were compared with IMT-GM-CSF-BCG (n = 10) and RFA alone (n = 44). PPD was the best candidate for BCG replacement.

For mice and pigs, 5 µg of mouse GM-CSF (mGM-CSF Miltenyi Biotec, Paris, France) or 250 µg of human GM-CSF (hGM-CSF; Partner Therapeutics, Inc, Lynnwood, WA, USA) and 2 or 5 IU of PPD (Tubertest^®^, Sanofi Pasteur, Lyon, France) were solubilized at 4 °C for 1 h in 50 µL or 10 mL of muco-adherent thermogel. Doses considered for humans have been used in pigs. IMT rheology characterization was based on prior studies [18,19]. A rheometer (Anton Paar model MCR 102, 911841 Courtaboeuf, France) was used to obtain the gelation temperature (Tg) of thermogel formulations. A volume of 750 µL of thermogel was loaded on the support, then the conical geometry mobile (diameter = 50 mm; angle = 1) was positioned at a distance of 0.1 mm from the support. Non-destructive oscillatory measurements at 1 Hz were applied, allowing us to obtain the elastic modulus (G′), the viscous modulus (G″), and the phase angle (tan σ = G″/G′). The plate was heated at a rate of 1 °C/min from 20 °C to 40 °C. The point at which G0 began to move to higher temperatures gave the sol–gel transition temperature. Each formulation was analyzed three times.

To monitor thermogel stability in mice using optical imaging, immunomodulators were replaced with cyanine-5-linked human serum albumin (Cyal-MT-gel) as well described in previous studies [20].

### 2.3. Imaging of Immunogel Stability

For MRI, a 7T spectrometer fitted with an ultra-shielded refrigerated 300 WB magnet (Bruker, Avance II, Evry, France) was equipped with a whole-body 40 mm birdcage RF coil (Bruker, Ullis, France) for mice and Paravision 5.1 acquisition software. A 2D multislice GE sequence (Hermitian pulse 3 ms, 30°, TR/TE 0.5 s/3 ms, 20 slices of thickness 1 mm, FOV = 3 × 3 cm) was used to determine the anatomical location of the tumor. Then, T2-weighted multiecho SE (RARE, 4 echo) images with fat suppression were recorded (TR = 3000 ms, TE = 40 ms, Hermitian pulse 2 ms, TR/TE 1.5 s/40 ms, RARE factor 4, with a slice thickness of 1 mm, FOV = 3 × 3 cm). Seven days after tumor implantation, Cyal–solution or Cyal–thermogel (60 µL) was injected into the tumor (n = 6). The gel was detected on MRI as a hypersignal, and the surface was obtained after analysis of the axial section. 

In parallel, CT26-Luc tumors were imaged using a charge-coupled device optical system (Photon IMAGER, Biospace Lab, Nesles-la-Vallée, France). Cyal-MT-gel or Cyal–solution mixture was injected into the tumor grafted in the flank of five mice for each group. Fluorescence signal was acquired with a CDDi camera (Biospace Lab, France, Nesles-la-Vallée, France) at different time points post injection. The quantification was performed over a 2 cm^2^ region of interest (ROI) and applied to the tumors using M3 vision software version 1.3.227123. Quantification of the kinetics of the fluorescence signal, calculated by the area under the curve as a function of time, was expressed as fluorescence.h^−1^. The release data were expressed as the percentage of the signal taken at t0 compared to the percentage of the signal at time t. Luminescence levels of the region of interest (ROI) were quantified in photons/s/sr standardized with the background signal. The ROI ratio of the signal in the tumor area has been normalized with the signal of the whole mouse at each time point, each mouse being its own control.

### 2.4. Radiofrequency Ablation–Intratumoral Immunotherapy in Mice

Thirty BCG-vaccinated female BALB/c mice, each bearing a 200 mm^3^ primary subcutaneous CT26 tumor, were randomly treated with RFA or RFA combined with synchronous IMT-GM-CSF-PPD injection using a Cool-tip™ RF generator (Radionics, Burlington, MA, USA) with 3-watt power source pulse energy delivery through a single 17-gauge electrode (Cool-tip, Covidien, Medtronic, Paris, France) easily inserted into the center of the subcutaneous tumor. A thermocouple at the electrode tip monitored the tissue temperature up to 60 °C to ensure complete ablation of the target lesion. After two cycles of RFA, 60 µL of IMT–gel was injected into the center of the treated tumor. Then, 2 × 10^5^ CT26 cells were injected into the opposite flank as secondary tumors. In the CT26 model, a subcutaneous tumor is the less favorable situation for the immune response [21].

To minimize the impact of recurrences and the impairment of immune response, related to the variance of the volume of the primary treated tumors, the RFA-GMCSF-PPD group (n: 15) was compared with all groups of RFA-treated mice (n: 59) including RFA-only treated mice in a preliminary study for PAMP selection. All RFAs were performed in the standard conditions previously described. 

The volume of secondary distant tumors was measured until day 26 with caliper according to classic formula V (mm^3^) = (width × length)^2^/2. Survival was monitored up to day 60.

### 2.5. Radiofrequency Ablation–Intralesional Immunotherapy in Pig Model

In six anesthetized female domestic pigs, a 3.5 cm × 15 cm RFA electrode (LeVeen™ CoAccess Needle ElectrodeM001262230; Boston Scientific, Voisin le Bretonneux, France) was placed percutaneously, through the CoAccess probe, into healthy liver tissue, guided by ultrasonography, as far as possible from the hepatic veins and gallbladder. Two RFA cycles per lesion, in the right and left lobes, were applied using a 480 kHz Boston Scientific RF 3000 generator with an impedance-based feedback algorithm. The power was started at 20 W and increased by 10 W/30 s (M.E.H. and R.M). Five minutes after ablation, 10 mL (pigs 1–2) or 5 mL (pigs 3–6) of IMT stored at 17 °C was injected into each lesion in one minute, through the CoAccess probe. The follow-up was conducted for 15 days. Liver ultrasonography and blood cell count, hematocrit, and liver tests (Cobas analyzer, Roche, Meylan, France) were performed on days 0, 7, and 15. Peripheral blood mononuclear cells were isolated [22], washed twice, and analyzed using flow cytometry (LSR Fortessa X20, Becton Dickinson, Erenbodegem, Belgium). Lymphocyte subsets were quantified using the following BD Biosciences antibodies: anti-pig PE-Cy7-CD3-(BB23-8E6-8C8), PerCP-Cy5.5-CD4-(74-12-4), FITC-CD8-(76-2-11), PE-TCR γδ-(MAC320), and Alexa-Fluor-647, CD29-(NaM160-1A3). 

An enzyme-linked immunosorbent assay (ELISA) kit (Thermo Fisher Scientific, Inc, Asnière, France) quantified hGM-CSF in pig sera, in triplicate, using a calibration curve. 

After euthanasia, the abdominal and thoracic organs were examined by the surgeon (R.M.) and the pathologist (J.F.E.) for signs of hemorrhage, infarction, or thermal injury. In liver and lung samples, formalin-fixed paraffin-embedded samples were used to obtain 4-micrometer thick sections that were stained with hematoxylin and eosin for histological analysis.

### 2.6. Transfemoral Thrombolysis

Before euthanizing pigs 4 and 6, the Seldinger technique was used to position a 6F sheath in the right femoral vein. An angiographic 5F JR4 catheter guided by a 0.035 wire was inserted inside the sheath into the suprahepatic inferior vena cava in pig 4 and the left pulmonary artery in pig 6. In pig 4, a 12 × 20 mm balloon (Mustang™ 7F; Boston Scientific, Voisin le Bretonneux, France) was positioned in the infra atrial vena cava, allowing us to obtain a venous thrombus without cardiac embolism by the injection of 5 mL of thermogel. In pig 6, 5 mL of thermogel was injected after catheterization of the left inferior pulmonary artery. Thrombolysis was performed with 150 mL of 10 °C saline serum using angiographic monitoring and thrombus imaging.

### 2.7. Statistical Analysis

The Mann–Whitney U test (GraphPad) was used to compare the differences between two groups. Two-way analysis of variance (ANOVA) with Tukey’s test was used for multiple comparisons. A log-rank test was used for survival analysis. Statistical significance was set at *p* < 0.05.

## 3. Results

### 3.1. Rheology and Imaging Validate Immune-Muco-Adherent Thermogel as a Reliable Intratumoral Immunotherapy Vector

The P407 21% XG 0.1% thermogel was chosen because of its basic rheological properties, as well described in previous publications and reinvestigated in the study combining two GM-CSF species. A sol–gel gelling temperature of 20–21 °C ensured easy injection of the gel at operative room temperature, and the elastic modulus value of G′ > 18,000 Pa at 37 °C confirmed the immediate gelling at body temperature, guaranteeing the local IMT stability (Table 1).

MRI (T2) detected the Cyal-MT-gel in the tumors for up to 72 h, which was not the case for the liquid solution of albumin–cyanine 5 (Cyal-Sol) (Figure 1a). Optical imaging allowed Cyal-MT-gel formulations to be followed for 7 days, which was not the case for liquid Cyal-Sol (Figure 1b). Quantification of the fluorescence signal was superior for the Cyal-MT-gel, with a value of 735.7 fluorescence.h^−1^ against 490.3 fluorescence.h^−1^ for Cyal-Sol at 168 h (Figure 1c). These data confirmed that the residence time of the thermogel in the RFA area was consistent with the time required to prime an effective anti-tumor immune response in situ.

### 3.2. Radiofrequency Ablation–Intratumoral Immunotherapy Primes an Abscopal Effect in Mice

To avoid the BCG infectious complications, PPD were selected (Appendix A) and evaluated, as shown in Appendix A and Figure 2a. In the CT26 model, treatment of primary tumors with RFA + gel-GM-CSF-PPD induced the regression of distant tumors and improved survival of mice compared to treatment by RFA alone (Table 2). In IMT-treated mice, the volume of secondary tumors was 106 ± 57 mm^3^ at day 16 and became undetectable at day 18. In the RFA group, the mean volume of distant lesions increased from 126.98 ± 27.83 mm^3^ at day 16 to 350 ± 99.3 mm^3^ at day 23, requiring euthanasia of 90% of the mice before day 26 (Figure 2b). On day 60, the survival rate of RFA-IMT-treated mice was 66% (10/15) compared to a rate of less than 10% in mice treated with RFA alone (5/59) (Figure 2c).

### 3.3. Radiofrequency Ablation–Intralesional Immunotherapy in Adjusted Volume Is Safe in Large Animals

In view of a phase I clinical trial, the feasibility and safety of the procedure must be assessed in conditions close to human anatomy. In the healthy livers of six pigs, 11 percutaneous RFA sites were injected with IMT–PPD (Table 3). No weight loss (Figure 3a) or clinical hemorrhagic, liver, or digestive disorders were observed during the 15 days of follow-up. No significant difference was detected between days 0 and 15 in terms of blood cells and platelet counts or hemoglobin level (Figure 3b–f). Bilirubin was within the range of the normal standard. Liver enzyme dosages showed a decrease in alkaline phosphatases (ALP) from 140.0 UI/L ± 12.5 on day 7 to 122.0 UI/L ± 9.8 on day 15 (*p* < 0.05); the increase in low transaminases (ALTP) from 63 UI/L ± 5.1 on day 0 to 91.4 UI/L ± 6.9 on day 15 (*p* < 0.05) (Figure 3g,h) may be related to RFA liver aggression.

Thermal lesions were characterized using perioperative US imaging as hyperechogenic areas. The IMT deposit was detectable as a hypoechogenic ring or area (Figure 4a–c). US measurements of RFA spots on days 7 and 14 confirmed the decrease in the lesions’ volume over time without detectable portal, biliary, or venous complications (Table 3).

During the second injection in pig 2, an acute refractory heart failure was associated with the migration of the thermogel into the cardiac cavities through the thermal rupture of a large hepatic vein (Figure 4a,d–f). The mechanism of this adverse event was suspected based on the US perioperative view, which showed close contact between the RFA lesion and a large hepatic vein (Figure 4a). Macroscopic and microscopic pathological examinations (Figure 4d–f) confirmed this hypothesis regarding the anatomical relationship between the thermal lesion and the hepatic vein lumen. A vein wall break-in and thrombi in contact with vein damage were observed. A reduction in the injection volume from 10 mL to 5 mL prevented this adverse event in the following animals. The other five animals remained hemodynamically stable during RFA-ILI treatment, with normal cardiac and O_2_ saturation parameters until recovery from anesthesia, and four were followed for 15 days (Table 3).

### 3.4. Radiofrequency Ablation–Intralesional Immunotherapy Changes T Cell Immunity Signal

Peripheral blood mononuclear cells (PBMCs) were isolated and analyzed using flow cytometry. Appendix A represent an example of the successive steps of PBMC characterization. RFA-ILI increased the proportion of CD3 T cells from 65.6% ± 6.2 on day 0 and 64.6 ± 5.3% on day 7 to 73.0 ± 5.7% (*p* < 0.01) 15 days after treatment. No significant changes were observed in the CD4 or CD8 levels. However, a decrease in CD29-expressing CD8 cells was detected, from 54.8 ± 6.8% on day 0 and 49.6 ± 2.1% on day 7 to 42.2 ± 5.1% on day 15 (*p* < 0.05), suggesting a decrease in cytotoxicity and inflammation, 2 weeks after treatment. The proportion of CD3 TCRγδ^+^ cells increased from 35.6 ± 2.5% on day 0 to 44.0 ± 7.3% on day 7 (*p* < 0.05) to 45.3 ± 7.1% on day 15 (*p* < 0.05) and remained stable between days 7 and 15 (Figure 3i). Very low hGM-CSF serum levels were consistently around the detection threshold for pig 1, treated with IMT without GM-CSF. GM-CSF was introduced in the thermogel injected into pigs 3–5. Release of hGM-CSF was not significantly detectable on day 7 (11.9 ± 6.5 pg/mL) and day 15 (9.9 ± 17.5 pg/mL) compared to the baseline on day 0 (12.1 ± 4.2 pg/mL) (Figure 3j).

### 3.5. Endovascular Rescue after Gel Extravasation

In two pigs, before euthanasia, an experimental gel thrombus was created and dissolved by infusion of serum at 10 °C into the suprahepatic inferior vena cava (pig 4) or left pulmonary artery (pig 6). Thrombus formation and thrombolysis were confirmed using fluoroscopic imaging. In the first animal (pig 4), the occlusive balloon prevented cardiac gel migration at the time of the injection. The vena cava thrombus was dissolved by infusion of 150 mL of 10 °C saline solution (Figure 5a–c). Cardiac dysfunction was not observed following balloon removal. Monitoring recorded a 10% transient desaturation and moderate bradycardia at 75 beats per minute (bpm) during cold vena cava infusion, followed by reactive tachycardia at 110 bpm, and a return to the baseline value of 95 bpm within 10 min. In pig 6, transfemoral catheterization of the left pulmonary artery was performed for gel embolization. The proximal pulmonary embolism was assessed using fluoroscopic imaging. During cold perfusion, anterograde recanalization was observed until complete radiological recovery of the pulmonary arterial tree was achieved within 7 min (Figure 5d–f).

## 4. Discussion

Oncolysis combined with intratumoral immunotherapy including a cytokine and TLR agonist could convert a distant, immunologically ‘cold’ tumor into an immunologically ‘hot’ tumor through a systemic immune response resulting in an abscopal effect transforming the TME [8,9,10,23].

PPD as a BCG replacement are effective, avoiding the risk of infectious complications. Systemic immune response and abscopal effect were confirmed in mice through the complete regression of distant lesions and 66% long-term survival in the RFA-ITI group vs. 10% after RFA. These results are consistent with those already published relating to BCG [10]. The efficacy of PPD in humans has been confirmed in melanoma [24] and for the intratumoral treatment of viral tumors as warts [25,26,27], for which studies have reported a significant increase in IL-4 and IL-12 serum levels, a complete response in 75% of patients, and no recurrence after intralesional PPD immunotherapy [14,28]. 

In micro metastatic situations, uncontrolled recurrence of the primary lesion is harmful to anti-tumor immunity [29]. In mice treated with PPD, low early mortality and long-lasting disease-free survival suggest a positive effect of ITI on the risk of local recurrence in the RFA area. PPD may have a greater availability for DCs within the gel structure compared to whole bacteria, as demonstrated by the greater efficiency of GM-CSF-gel compared to whole bacterial gel in maturation of DCs [19]. PPD activate Th1, CD4 T, and NK cells, inducing local cytolytic activity, which may improve local control of the primary tumor [30,31]. A direct effect of PPD reverses the deactivation of monocytes related to contact with tumor cells [32] and can promote the production of Th1-related cytokines when it is combined with the destruction of tumor cells expressing TGF-β [33]. These local effects could overcome the limitations of RFA by decreasing the risk of perilesional recurrence related to tumor size or vascular-mediated cooling, which is known as the heat-sink effect.

In pigs, we observed a significant increase in the proportion of CD3 T cells, particularly CD3 TCRγδ^+^ T cells, between day 0 and days 7–14. Unlike mice and humans, porcine CD3 TCRγδ^+^ T cells represent a prominent subset of T cells in the blood and secondary lymphatic organs and contribute to immune responses in a unique way [34]. The integrin β1 chain (CD29) is a target of natural antibodies involved in the rejection of pig-to-human xenografts [35] and identifies cytotoxic CD4 and CD8 T cells [36,37]. The significant decrease in CD8 T cells expressing CD29 at D15 suggests a decrease in cytotoxicity and inflammation 2 weeks after treatment. Compared to baseline levels, the systemic release of hGM-CSF was undetectable. The measured plasma concentrations were comparable to the serum concentrations in patients treated with safe GM-CSF-secreting vaccines such as GVAX [38]. This result is a strong argument to explain the absence of side event, particularly immunosuppressive effects, which have been described in high serum concentrations of GM-CSF [39,40].

Complications related to RFA vessel damage include Budd–Chiari syndrome and pulmonary embolism [41]. Anatomical limitations explain the difficulties of intrahepatic RFA-ILI evaluation in pig model. The liver size in young specimens is small, less than 900 mL, and contains four large hepatic veins with thinner walls than those in humans [42]. The vascular breaking-in and extrahepatic gel migration in pig 2 may be related to the vein proximity and fragility. In a healthy liver, unlike in tumors and cirrhotic livers, the absence of a desmoplastic reaction, fibrosis, and pseudo-capsules increases energy transmission and the risk of vascular injury. The heat-sink effect, which downgrades thermocoagulation, is reduced during RFA in healthy livers owing to enhanced conductivity in normal tissues. These conditions increase the risk of large hepatic vein trauma. In humans, a volume of 5 mL/lesion can be selected because the entire dose of immunomodulators can be preserved by increasing the concentration. Moreover, the anatomical conditions, preoperative selection of target lesions, and control of hepatic venous pressure can prevent gel migration.

In this context, we demonstrated for the first time the feasibility of pulmonary artery thermogel thrombolysis. In the case of hepatic vein break-in and gel migration, the in situ perfusion of cold serum allows complete and definitive thrombolysis in less than 10 min.

Differential diagnosis of gel migration is the complement activation-related pseudo-allergy (CARPA), an acute hypersensitivity to allergens, drugs, and drug carriers, for which the pig is a sensitive model [43]. It involves abnormalities in the cardiac electrical conductance and ventricular function related to the activation of mast cells in the heart involving C5a rather than the C3a complement fraction [44]. In our study, embolism in pig 2 was established based on suddenness of cardiac arrest, without previous clinical manifestations of hypersensitivity such as tachycardia, paradoxical bradycardia, ECG abnormalities, bronchospasm, skin flushing, or rash, [44,45]. This diagnosis was mainly based on pathologic examination, which confirmed the hepatic vein break-in. There were no biological signs of hypersensitivity, particularly anemia, leuko-thrombocytopenia, leukocytosis, or shock aftermath, such as liver dysfunction, in any of the other animals. Reliability of postmortem biological analysis is precluded by the massive hemolysis. The strongest argument against the diagnosis of CARPA was the absence of symptoms after thrombolysis in the vena cava or pulmonary artery in animals 4 and 6.

RFA-ITI using PPD is effective in mice in treated CRC lesions and distant micro metastases that cause recurrences after RFA. This strategy, never validated in humans, uses a mini-invasive interventional radiologic procedure and a safe intratumoral immunotherapy, for which the absence of “out target” effect was shown [8,46]. This “in one go’’ interventional oncology, compatible with factual treatments, would improve survival and quality of life of patients with colorectal liver metastases.

## Figures and Tables

**Figure 1 pharmaceutics-16-00315-f001:**
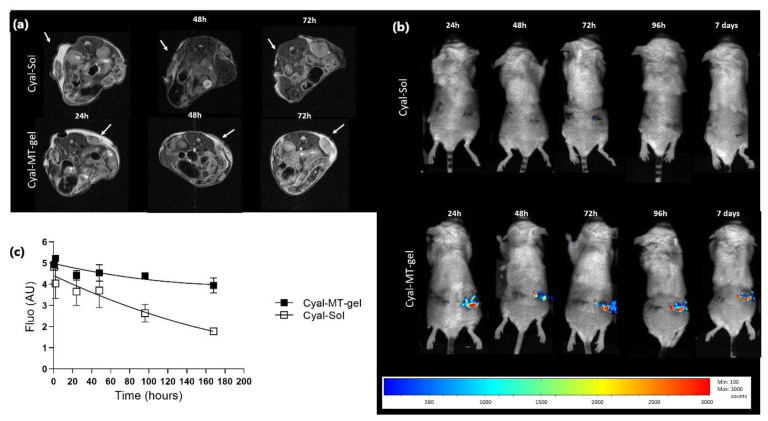
Imaging of gel stability in six BALB/c mice bearing a CT26-Luc tumor in flank with cyanine 5 solution (Cyal-Sol) in top pictures and cyanine 5 in muco-adhesive thermogel (Cyal-MT-gel) in lower pictures; (**a**) MRI T2-weighted multiecho SE images with fat suppression. The temperature of each animal was monitored. Breath triggering and respiratory gating were monitored (SA Instruments Inc., New York, NY, USA). The detection of the hypersignal related to the water restricted in the gel was considered as the localization of the thermogel. The images obtained at 24, 48, and 72 h show the effect of the adhesive properties of the gel. The hypersignal was detectable in the Cyal-MT-gel-injected tumor area for up to 72 h. Cyal-Sol signaling was not detectable after 24 h. (**b**) shows a representative mouse from each of the Cyal-MT-gel or Cyal-Sol groups and the evolution of the injection of cyanine-5-labeled gel and solution until the 7th day. The injected tumor signal intensity is represented in color from low intensity in blue to high intensity in red. Quantification of the kinetics of the luminescence signal, calculated by the area under the curve as a function of time, is expressed as luminescence.h^−1^. The ROI ratio of the signal in the tumor area has been normalized with the signal of the whole mouse at each time point, each mouse being its own control. The Cyal-Sol signal was not visible after 24 h. Cyal-MT-gel was present on day 7 with the same qualitative intensity from days 1 to 7. (**c**) Decrease in gel luminescence signal in tumor areas measured until 168 h and expressed as the percentage of the signal taken at t0 compared to the percentage of the signal at time t.

**Figure 2 pharmaceutics-16-00315-f002:**
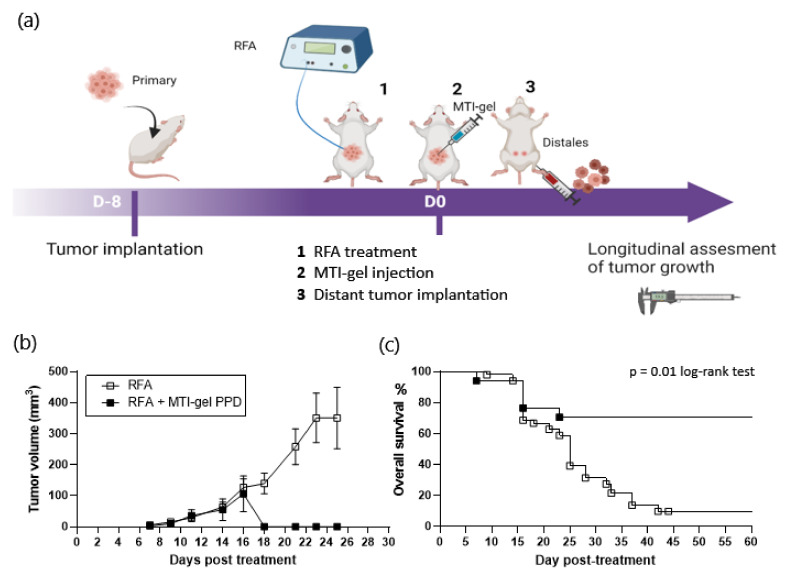
Experimental design and results of the RFA (n = 44 + 15) and RFA–intratumoral immunotherapy combining immune-muco-adherent thermogel (IMT), granulocyte–macrophage colony-stimulating factor (GMCSF), and purified protein derivatives (PPD) (IMT-gel PPD) (n = 15) on distant, untreated CT26 tumors in BALB/c mice. To minimize the effects of variability in the volume and recurrence of the primary treated tumor, all groups of standard RFA-treated mice (n = 44) in the preliminary experiments were used for the final evaluation of IMT-PPD in the study; (**a**) schematic representation of the timeline of treatment in CT26 murine model. Longitudinal measurement of tumor volume was recorded every 2 days until day 26, and survival curve was completed on day 60; (**b**) volume of subcutaneous secondary lesion. The data are presented as mean ± SEM. Increase in secondary tumor volume was comparable in the RFA and RFA-IMT groups on day 16. Between day 16 and day 18, secondary tumors became unmeasurable in surviving mice treated by RFA-IMT-gel PPD while tumor growth in mice treated by RFA alone increased, reaching 350 ± 99.3 mm^3^ at day 25 (sacrifice at this limit point); (**c**) survival of mice was recorded on day 60. Curves reflect the abscopal effect of intratumoral immunotherapy on secondary tumors. After regression of the secondary lesions, 90% of the mice were alive without recurrence until day 60. In the RFA group, the survival rate decreased to less than 10% on day 60 (sacrifice at the limit point), *p* = 0.01.

**Figure 3 pharmaceutics-16-00315-f003:**
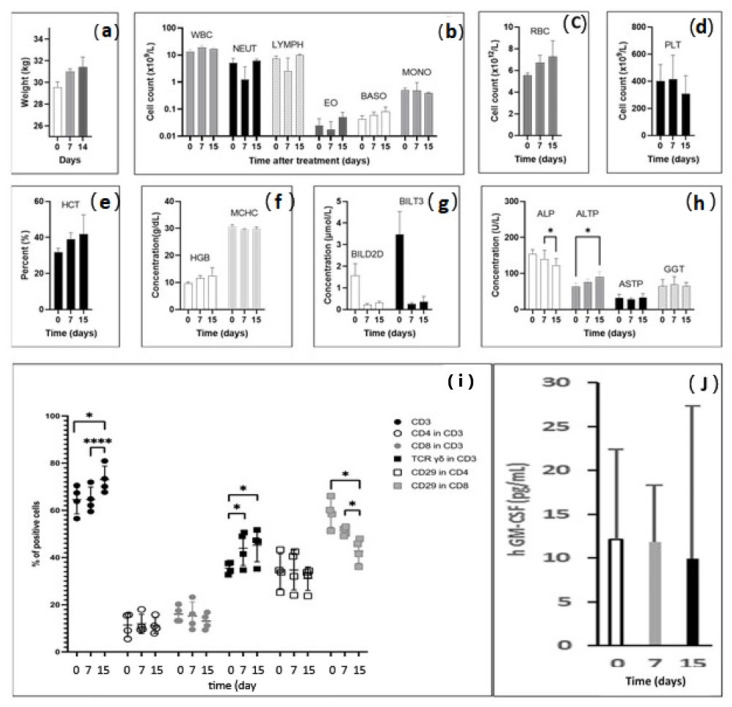
Clinical, biological, and immunological parameters after RFA intralesional immunotherapy in pigs; data are in means ± standard deviations; (**a**) body weight of pigs increased from 29.2 kg at baseline to 30.8 kg on day 7 and to 31.4 kg on day 15; (**b**–**d**) blood cell counts, white blood cells (WBC), neutrophils (NEU), red blood cells (RBC), and platelets (PLT) were not altered after radiofrequency intralesional immunotherapy; (**e**,**f**) hematocrit and hemoglobin levels remained stable within the normal range; (**g**) Bilirubin levels were higher at baseline but within the normal range (<5 µmol/L); (**h**) liver enzyme dosages showed a decrease in alkaline phosphatases (ALP) between day 7 and day 15 (* *p* < 0.05), and transaminases (ALTP) increased from day 0 to day 15 (* *p* < 0.05); (**i**) fluorescence-activated cell sorting (FACS) lymphocyte characterization. RFA-ITI increased the total proportion of CD3T cells from day 0 and from day 7 to day 15 (**** *p* < 0.01). Proportions of CD4 and CD8 cells were comparable at each time point. Proportion of CD29-expressing CD8 cells decreased from day 0 and day 7 to day 15 (* *p* < 0.05). The proportion of CD3 TCRγδ+ increased from day 0 and day 7 (* *p* < 0.05) and day 15 (* *p* < 0.05); (**j**) human GM-CSF serum levels determined by enzyme-linked immunosorbent assay (ELISA). The limit of detection of human GM-CSF was 2.9 pg/mL. Systemic release of GM-CSF was not significantly detectable.

**Figure 4 pharmaceutics-16-00315-f004:**
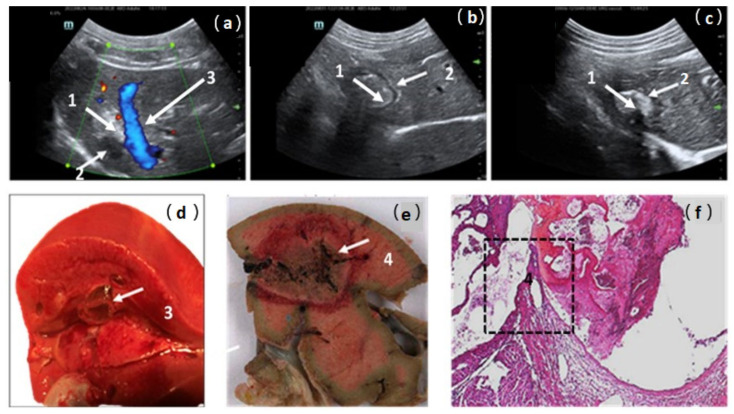
Ultrasonographic post-operative views of treated area in pig liver; (**a**) day 0; (**b**) day 7; (**c**) day 15. RFA-treated area was visible as a hyperechogenic lesion (1) compared to healthy liver. Intralesional immunotherapy imaging was detectable as a hypoechogenic area or ring (2). In pig 2, the close contact between the liver RFA area and a large hepatic vein (3) explains the possibility of vessel damage and the risk of gel migration; (**d**,**e**) macroscopic views confirm the relationship between RFA area and the large-caliber hepatic vein, the wall break-in particularly (4); (**f**) four-micrometer-thick sections of formalin-fixed paraffin-embedded samples stained by hematoxylin–eosin. Original magnification (×40) views confirmed, in the black box, the vein wall break-in and a recent thrombus in close contact with the area of thermocoagulation (4).

**Figure 5 pharmaceutics-16-00315-f005:**
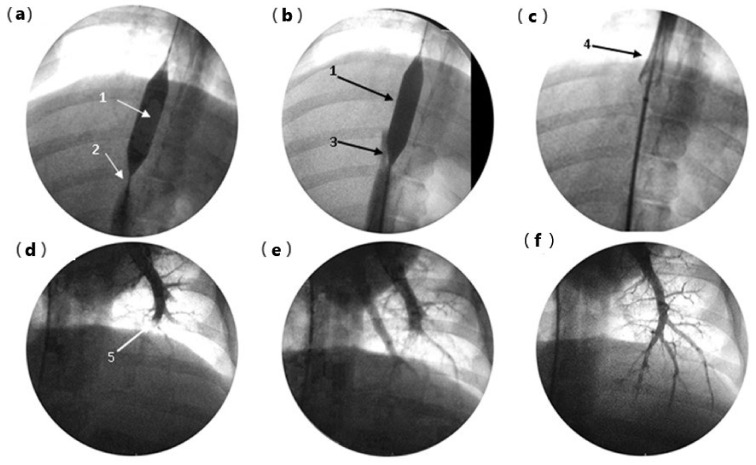
Endovascular rescue after gel embolism: (**a**) pig 4 embolization of the retro hepatic vena cava (RHVC), cardio protection by a 12 × 20 mm balloon (1), 10 mL of gel-forming thrombus in the inferior vena cava, upstream of the balloon, visible as a subtraction image (2); (**b**) start of recanalization during 10 °C serum perfusion, visible as progression of the contrast around the balloon (3); (**c**) deflation of balloon and repermeabilization of RHVC completed without cardiopulmonary complication (4); (**d**) pig 6: left pulmonary artery (PA) gel embolization (5); (**e**) proximal repermeabilization after 10 °C serum perfusion, time = 3 min; (**f**) complete recovery of PA tree after 10 °C perfusion, time = 7 min.

**Table 1 pharmaceutics-16-00315-t001:** Rheological characterization of thermogels.

	T°	G′ (Pa) *	G″ (Pa) *	T Sol–Gel (°C) *
P407 20% XG 0.1%	15	1.03	0.97	21.1
	40	21,209	566	
P407 20% XG 0.1% + mGM-CSF	15	1.69	0.90	18.8
	40	25,865	390	
P407 20% XG 0.1% + hGM-CSF	15	7.00	0.89	21.1
	40	18,269	879	

P407, poloxamer, XG, xanthan gum, mGM-CSF, mouse granulocyte–macrophage colony-stimulating factor, hGM-CSF, human GM-CSF, G′, elastic modulus, G″, viscous modulus, T Sol–Gel, gelling temperature. * Rheometer (MCR 102; Anton Paar, Les Ulis, France) was used to determine the gelation temperature (T Sol–Gel). Oscillatory measurements were performed at 1 Hz to obtain the elastic modulus (G′) and viscous modulus (G″).

**Table 2 pharmaceutics-16-00315-t002:** Survival and secondary tumor volume in mice.

RFA			RFA-IMT PPD	
**n**	**Mean**	**Day**	**Mean**	**n**
52 *	5.71 ± 2.31	7	2.61 ± 1.48	14 *
52	15.25 ± 3.88	9	9.80 ± 5.59	14
51	29.30 ± 6.96	11	35.74 ± 19.54	14
49	63.90 ± 16.08	14	54.82 ± 34.64	14
45	126.98 ± 27.83	16	106.14 ± 57.87	11
40	139.32 ± 33.06	18	0.00	11
39	258.09 ± 57.64	21	0.00	11
37	350.93 ± 79.62 †	23	0.00	10
31	350.65 + 99.29 †	25	0.00	10

Secondary tumors mean volume in mm^3^, RFA, radiofrequency, IMT, immune-muco-adherent thermogel, PPD, purified protein derivatives. * On day 7, perioperative mortality was excluded. † A volume greater than 350 mm^3^ was considered the limit point for euthanasia.

**Table 3 pharmaceutics-16-00315-t003:** Characteristics of liver radiofrequency immune-muco-adherent thermogel injection and parameters in pigs.

No.		1	2	3	4	5	6
RFA		Two cycles	Two cycles	Two cycles	Two cycles	Two cycles	Two cycles
Sites		1	2	2	2	2	2
Gel injection		10 mL	10/10 mL	5/3 mL	5/5 mL	5/5 mL	5/5 mL
Immuno		PPD ^§^	PPD ^§^	PPD hGMCSF	PPD hGMCSF	PPD hGMCSF	-
Mean CF (beats/mn)		94 ± 1.0		88 ± 4.2	95 ± 0.9	81 ± 3.3	
Mean SPO_2_ (%)		96 ± 1.5		94 ± 0.3	95 ± 0.3	93 ± 0.7	
Follow-up		14 days	-	14 days	14 days	14 days	-
Additional process					➢ Femoral catheterism; ➢ Occlusion by gel injection in supra hepatic vena cava; ➢ Thrombolysis: 150 mL of 10 °C saline serum perfusion.		➢ Femoral catheterism; ➢ Occlusion by gel injection of left pulmonary artery; ➢ Thrombolysis: 150 mL of 10 °C saline serum perfusion.
							mean ± SD
Liver lesions * (mm^3^)	d 0	90	157–113	89–100	88–75	137–65	92 ± 9
	d 7	89	-	77–80	65–62	138–61	82 ± 10
	d 15	60	-	79–61	62–50	104–55	67 ± 9

Data are means ± standard deviations. RFA, radiofrequency ablation, PPD, purified protein derivatives, hGM-CSF, human granulocyte–macrophage colony-stimulating factor, CF, cardiac frequency, SPO_2_, oxygen saturation. ^§^ Pigs treated without GMCSF as control for ELISA evaluation of the cytokine release. * Volume of RFA lesions was measured by ultrasonography on days 0, 7, and 15.

## Data Availability

The data presented in this study are available in this article (and Appendix A).

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
