# Peer review of "Radiofrequency Combined with Intratumoral Immunotherapy: Preclinical Results and Safety in Metastatic Colorectal Carcinoma"

_pharmaceutics, 2024, doi:10.3390/pharmaceutics16030315_

Round 1
Reviewer 1 Report
Comments and Suggestions for Authors
The manuscript by Seguin and Hajjam et al. “Radiofrequency Combined to Intratumoral Immunotherapy: Preclinical Results and Safety in Metastatic Colorectal Carcinoma” reports preclinical efficacy of combination therapy of RFA-ITI for colorectal cancer as well as biological safety of a thermogel in a large animal. RFA is a widely used thermal ablation therapeutic modality for liver cancer, yet ineffective in preventing tumor reoccurrence. The hypothesis to improve its efficacy through combination of immune stimulation is highly valid and relevant, yet the experimental process to test this hypothesis lacks rigor in this manuscript. Aside from subcutaneous, not orthotopic tumor growth, it is not clear whether the thermogel releasing GM-CSF and PPD achieved intra-tumor vaccination as it was not shown nor discussed that this immunostimulatory process would have had failed to control distal tumor growth if implanted somewhere else, particularly with the concern of gel-induced embolism.
Comments on the Quality of English LanguageMinor mistakes include line 26, “prevent” should be “preventing”; line 94 “establish” should be “establishing”; line 155, “2.105” should be “2x105”.
Author Response
The manuscript by Seguin and Hajjam et al. “Radiofrequency Combined to Intratumoral Immunotherapy: Preclinical Results and Safety in Metastatic Colorectal Carcinoma” reports preclinical efficacy of combination therapy of RFA-ITI for colorectal cancer as well as biological safety of a thermogel in a large animal. RFA is a widely used thermal ablation therapeutic modality for liver cancer, yet ineffective in preventing tumor reoccurrence. The hypothesis to improve its efficacy through combination of immune stimulation is highly valid and relevant, yet the experimental process to test this hypothesis lacks rigor in this manuscript.
- Aside from subcutaneous, not orthotopic tumor growth:
- CT 26 is a rectal tumor RFA can’t be performed in orthotopic situation
- RFA is very harmful in a 30g mouse, mortality is very high elsewhere than in a subcutaneous situation. This choice is also in compliance with “ Reduction” in 3R ethical recommendations
- it is not clear whether the thermogel releasing GM-CSF and PPD achieved intra-tumor vaccination as it was not shown nor discussed that this immunostimulatory process would have had failed to control distal tumor growth if implanted somewhere else:
- In mice characterization of tumor growth is more difficult, in deep situations. Direct measurement is impossible without sacrificing the animal and measurement by imaging or bioluminescence is aggressive for small animals, requires iterative anesthesia, and is imprecise, especially for small tumor volumes.
- In CT26 model, subcutaneous tumor is the less favorable situation for the immune response. This disadvantage therefore supports the significance of the results. This argument is now clarified and referenced in the text (ligne 157-158
Zhao, X.; Li, L.; Starr, T.K.; Subramanian, S. Tumor Location Impacts Immune Response in Mouse Models of Colon Cancer. Oncotarget 2017, 8, 54775–54787, doi:10.18632/oncotarget.18423.
- particularly with the concern of gel-induced embolism
- Gel embolism has only been performed in pigs; it is not possible in mouse. The objective of this experiment was only to anticipate the intra-operative risk in humans in a large size model.
Comments on the Quality of English Language
Minor mistakes include:
- line 26, “prevent” should be “preventing”: corrected
- line 94 “establish” should be “establishing”: corrected
- line 155, “2.105”should be “2x105”: corrected
Reviewer 2 Report
Comments and Suggestions for Authors
Johanne Seguin et al presented a combination therapy of radiofrequency and intratumoral immunotherapy via immune-muco-adherent thermogel and evaluated its efficacy in a murine metastatic colorectal tumor model and safety in a pig model. The manuscript is well-structured while the quality of English language should be improved. Besides, please address the following concerns to improve the quality of the manuscript.
1. Line 41: To compare RFA-IMT group with RFA group, the author should compare the distant lesions on same day (either day 16 or day 25). Based on Line 245, the regression of distant lesions was not “completed” on “day 16”, but “day 18”. Moreover, the phrase “by day 16 vs a volume of 350 + 99.3 mm^3” could be clearer. For example, “by day 16, as opposed to a distant tumor volume of 350 ± 99.3 mm^3 in the RFA group". Similarly, please modify Line 246 and change “350 mm^3± 99.3” to “350 ± 99.3 mm^3”.
2. The author should update Figure 2 to clearly indicate how the 44 mice are treated differently. From Figure S1 caption, the total number of mice is 10 (RFA) + 10 (RFA-PPD)+10 (RFA-BCG)+11(RFA-MEPACT) +4 (RFA-MPLA) = 45. And if that’s the case, the tumor growth results from BCG/ MEPACT /MPLA groups shouldn’t be included.
3. Figure S1a&d. Only 2 curves are recognizable. Hard to distinguish “Control” with “RFA”. The resolution of the figure should be improved.
4. Figure 3b, please present primary tumor volume data as well.
Comments on the Quality of English Language1. Line 25: “inadequate for prevent recurrence” should be “insufficient to prevent recurrence”.
2. Use “in mice” or “in murine models” instead of “in mouse”. Same rule applies to “pig”.
3. Be consistent with acronyms. “Cyal MT gel” (Line 138), “Cyal-MT Gel” (Line 226), “Cyal-MT-gel” (Line 216) are used.
4. Figure 3a, “Distant tumor implantation” instead of “Distales tumors injection”.
5. Line 228: “Breath triggering was and respiratory gating was monitored” should be “Breath triggering and respiratory gating were monitored”.
Author Response
Johanne Seguin et al presented a combination therapy of radiofrequency and intratumoral immunotherapy via immune-muco-adherent thermogel and evaluated its efficacy in a murine metastatic colorectal tumor model and safety in a pig model. The manuscript is well-structured while the quality of English language should be improved. Besides, please address the following concerns to improve the quality of the manuscript.
- Line 41: To compare RFA-IMT group with RFA group, the author should compare the distant lesions on same day (either day 16 or day 25). Based on Line 245, the regression of distant lesions was not “completed” on “day 16”, but “day 18”. Moreover, the phrase “by day 16 vs a volume of 350 + 99.3 mm^3” could be clearer. For example, “by day 16, as opposed to a distant tumor volume of 350 ± 99.3 mm^3 in the RFA group". Similarly, please modify Line 246 and change “350 mm^3± 99.3” to “350 ± 99.3 mm^3”:
- Distant lesions was cured from day 18 in RFA-IMT group and remained undetectable up to day 60 while in the RFA group the lesions continued their exponential growth up to the limit point from day 23 to day 26 (corrected in text).
- The kinetic of distant tumor regression in the PPD group was strictly comparable with the kinetic of tumor regression in mice treated by BCG extensively described in our previous publication [ ref 10]
- Line 246 corrected
- The author should update Figure 2 to clearly indicate how the 44 mice are treated differently. From Figure S1 caption, the total number of mice is 10 (RFA) + 10 (RFA-PPD)+10 (RFA-BCG)+11(RFA-MEPACT) +4 (RFA-MPLA) = 45. And if that’s the case, the tumor growth results from BCG/ MEPACT /MPLA groups shouldn’t be included:
- Figure S1a&d. Only 2 curves are recognizable. Hard to distinguish “Control” with “RFA”. The resolution of the figure should be improved.
- Indeed Figure S1 was confusing and unclear and the PAMPs selection was not the objective of the study. The tumor growth results from BCG/ MEPACT /MPLA groups are canceled. As recommended by third reviewer this figure was replaced as supplementary by figure 2 clarifying more simply the improvement of the size of the sample of RFA alone treated mice. In the preliminary experiments, which selected the PPD among other PAMP candidates, 44 mice were treated by RFA alone under strictly controlled conditions. These animals were included in the study to increase the power of the study (in decreasing the impact of the size of the primary tumor at the time of RFA, by example). These 44 mice were treated by RFA alone were treated in exactly the same way as the 15 mice included in the RFA group of the study.
- Figure 3b, please present primary tumor volume data as well.
- The volume of the primary tumor destroyed by RFA was not measurable in the absence of recurrence (the more frequent event), or it was very difficult to measure reccurence with precision in the inflamed fibrosis of the RFA scar.
Comments on the Quality of English Language
- Line 25: “inadequate for prevent recurrence” should be “insufficient to prevent recurrence”: corrected
- Use “in mice” or “in murine models” instead of “in mouse”. Same rule applies to “pig”: corrected
- 3. Be consistent with acronyms. “Cyal MT gel” (Line 138), “Cyal-MT Gel” (Line 226), “Cyal-MT-gel” (Line 216) are used: corrected
- Figure 3a, “Distant tumor implantation” instead of “Distales tumors injection”: corrected
- Line 228: “Breath triggering was and respiratory gating was monitored” should be “Breath triggering and respiratory gating were monitored”: corrected
Reviewer 3 Report
Comments and Suggestions for Authors
In this study the authors examined efficiency of RFA-intratumoral immunotherapy using GM-CSF and PPD combined in a murine and pig model. The study holds significant amount of preclinical data, which could lead to colorectal cancer therapy improvement. Moreover, the conclusions are supported with in vitro and animal model data. Overall, the manuscript is well prepared, the methods are described in detail and the study is important.
The concerns:
1) Please clarify what was the baseline fluorescence in mice tumors without Cyal-Sol or Cyal-MT-gel (figure 1b/c)?
2) The tumor volume decrease in RFA+MTI-gel PPD between day 16 and day 18 is astonishing! Could the authors provide additional evidence for such significant tumor reduction (e.g. validate histologically)? Could this data have biased by measure collecting technique?
3) Why didn’t the authors evaluate blood cell characteristics in mice? Please explain.
4) Control group for the injection with IMT—PPD is missing. Please explain.
Minor:
1) The authors could enhance the quality of the figures;
2) Please combine figure 2 with figure 3 or place it as supplementary;
3) Please depict value of statistical difference in figure 3c;
4) Figure 6 could be placed as supplementary.
Author Response
In this study the authors examined efficiency of RFA-intratumoral immunotherapy using GM-CSF and PPD combined in a murine and pig model. The study holds significant amount of preclinical data, which could lead to colorectal cancer therapy improvement. Moreover, the conclusions are supported with in vitro and animal model data. Overall, the manuscript is well prepared, the methods are described in detail and the study is important.
The concerns:
- Please clarify what was the baseline fluorescence in mice tumors without Cyal-Sol or Cyal-MT-gel (figure 1b/c)?
- To prove the higher residence time of Cyal-MT-gel, we followed fluorescence versus time after injection. Figure 1b shows a representative mouse from each Cyal-MT-gel or Cyal-sol group. If we plot a region of interest in non-injected mice, data not shown, we obtain a fluorescence value of 48 +/-119 counts, which is approximately the level of the plate background 52+/-103. In terms of comparison, the 24h signal value for Cyal-MT-gel mice it is 1830000 counts. In Figure 1c, the ROI ratio of the signal in the tumor area has been normalized with the signal of the whole mouse at each time point, each mouse being its own control. This point is included in lines 147-148 and in legend of figure 1.
- The tumor volume decrease in RFA+MTI-gel PPD between day 16 and day 18 is astonishing! Could the authors provide additional evidence for such significant tumor reduction (e.g. validate histologically)? Could this data have biased by measure collecting technique?
- In this study, whose objective was to verify the preclinical effectiveness of replacing BCG with PPD, we found exactly the same results on tumor volume as in our previous work performed with BCG, already published and cited in reference [ref 10]. The tumor volume also decreased in RFA+MTI-gel BCG between day 16 and day 18. This kinetic of tumor regression was also confirmed in bioluminescence.
- Mechanisms of such tumor regression (histological, and T cell response) are explained extensively in our previous work cited in reference [ref10].
- The effect on survival at day 60 confirm the reality of tumor regression. In CT26 model, any mouse is alive at day 60 with a progressive tumor.
- Reference of these data is introduced in line 386
3) Why didn’t the authors evaluate blood cell characteristics in mice? Please explain
These parameters were extensively detailed in our previous works [Ref 10 and 19]
The objective of this study was to assess the preclinical effectiveness of replacing alive BCG with safe PPD. With this preclinical view, the two more important endpoints in mice was volumetric tumor regression and survival. The clinical response to PPD-intratumoral immunotherapy is strictly modeled on response with BCG. With adding the T cell response analysis, that we have already dissected and published, it would not possible to group, in one publication, the preclinical explorations in 2 animal models, useful in terms of effectiveness and safety.-
- Control group for the injection with IMT—PPD is missing. Please explain.
The control group in our preclinical study is represented, as in a perspective of a clinical trial, by the mice treated by RFA alone. The evolution of a control group of mice untreated or mistreated is constant and well described in our previous works [ref 10, 19]. In the untreated mice, the median tumor volume reach the limit point 235.1 mm3 at day 15. This group needs early euthanasia, is not strictly useful and it reduces compliance with “Reduction” in 3R ethical recommendations. An untreated group of patient is unrealistic in a clinical perspective.
Minor:
1) The authors could enhance the quality of the figures: the figure S1 with a major problem of visibility is canceled. The resolution of other figures is fixed by the device or the software of images acquisition. Modification of resolution is possible but needs manipulation of the image files, the modification in format and finally downgrades the readability.
2) Please combine figure 2 with figure 3 or place it as supplementary: figure 2 is now placed as supplementary
3) Please depict value of statistical difference in figure 3c;
We perform a log-rank test on the survival data of the two groups of mice. We want to calculate the pvalue of the hypothesis H0 (where we assume that the two groups are statistically identical).
Under H0, the statistic
Oi is the number of observed events for group i. Ei is the expected number of events for group I follows a χ2 distribution with one degree of freedom. It represents the distance between the observed events and the number of events that we should have obtained if the two groups were statistically identical.
To calculate Oi, it is sufficient to count the number of dead mice in the group i at the end of the study. The quantity Ei is calculated by summing the theoretical number of events that should have occurred in each group at each time increment of the study. The expected number of events in group i at time j is calculated as follows:
where dj is the total number of events at time j, rj is the total number of living mice at time j, and rij is the number of living mice in group i at time j. We note that Eij corresponds well to the expected number of events at time j for group i if the two groups were statistically identical (We calculate the overall probability of having an event at time j and multiply it by the number of living mice of group i at that time). Thus:
We find E1 = 39.3, E2 = 11.7, O1 = 47, O2 = 4. Thus, T = 6.58 therefore:
p value = 0.010313
included in statistical analysis chapter and in figure 2
4) Figure 6 could be placed as supplementary: now placed as supplementary
Round 2
Reviewer 1 Report
Comments and Suggestions for Authors
No more comments.
Reviewer 3 Report
Comments and Suggestions for Authors
The authors clarified all my concerns.